# Studying Lexical Dynamics and Language Change via Generalized Entropies: The Problem of Sample Size

**DOI:** 10.3390/e21050464

**Published:** 2019-05-03

**Authors:** Alexander Koplenig, Sascha Wolfer, Carolin Müller-Spitzer

**Affiliations:** Department of Lexical Studies, Institute for the German language (IDS), 68161 Mannheim, Germany

**Keywords:** generalized entropy, generalized divergence, Jensen–Shannon divergence, sample size, text length, Zipf’s law

## Abstract

Recently, it was demonstrated that generalized entropies of order α offer novel and important opportunities to quantify the similarity of symbol sequences where α is a free parameter. Varying this parameter makes it possible to magnify differences between different texts at specific scales of the corresponding word frequency spectrum. For the analysis of the statistical properties of natural languages, this is especially interesting, because textual data are characterized by Zipf’s law, i.e., there are very few word types that occur very often (e.g., function words expressing grammatical relationships) and many word types with a very low frequency (e.g., content words carrying most of the meaning of a sentence). Here, this approach is systematically and empirically studied by analyzing the lexical dynamics of the German weekly news magazine *Der Spiegel* (consisting of approximately 365,000 articles and 237,000,000 words that were published between 1947 and 2017). We show that, analogous to most other measures in quantitative linguistics, similarity measures based on generalized entropies depend heavily on the sample size (i.e., text length). We argue that this makes it difficult to quantify lexical dynamics and language change and show that standard sampling approaches do not solve this problem. We discuss the consequences of the results for the statistical analysis of languages.

## 1. Introduction

At a very basic level, the quantitative study of natural languages is about counting words: if a word occurs very often in one text but not in a second one, then we conclude that this difference might have some kind of significance for classifying both texts [1]. If a word occurs very often after another word, then we conclude that this might have some kind of significance in speech and language processing [2]. In both examples, we can use the gained knowledge to make informed predictions “with accuracy better than chance” [3], thus leading us to information theory quite naturally. If we consider each word type *i* = 1, 2, …, *K* as one distinct symbol, then we can count how often each word type appears in a document or text *t* and call the resulting word token frequency $f_{i}$. We can then represent *t* as a distribution of word frequencies. In order to quantify the amount of information contained in *t*, we can calculate the Gibbs–Shannon entropy of this distribution as [4]:(1)$$H\left(p\right)=-\overset{K}{\underset{i=1}{\sum }}p_{i}*log_{2}\left(p_{i}\right)$$
where $p_{i}=\frac{f_{i}}{N}$ is the maximum likelihood estimator of the probability of *i* in *t* for a database of *N* = $\sum_{i=1}^{K}f_{i}$ tokens. In [5], word entropies are estimated for more than 1000 languages. The results are then interpreted in light of information-theoretic models of communication, in which it is argued that word entropy constitutes a basic property of natural languages. *H*(*p*) can be interpreted as the average number of guesses required to correctly predict the type of word token that is randomly sampled from the entire text base (more precisely, [4], Section 5.7) show that the expected number of guesses *EG* satisfies *H*(*p*) ≤ *EG* < *H*(*p*) + 1). In the present paper, we analyze the lexical dynamics of the German weekly news magazine *Der Spiegel* (consisting of *N* = 236,743,042 word tokens, *K* = 4,009,318 different word types, and 365,514 articles that were published between 1947 and 2017; details on the database and preprocessing are presented Section 2). If the only knowledge we possess about the database were *K*, the number of different word types, then we would need on average *H*_max_ = log_2_(*K*) = log_2_(4,009,318) ≈ 21.93 guesses to correctly predict the word type, calculating *H* for our database based on Equation (1) using the corresponding probabilities for each *i* yields 12.28. The difference between *H*_max_ and H(p) is defined as information in [3]. Thus, knowledge of the non-uniform word frequency distribution gives us approximately 9.65 bits of information, or put differently, we save on average almost 10 guesses to correctly predict the word type.

To quantify the (dis)similarity between two different texts or databases, word entropies can be used to calculate the so-called Jensen–Shannon divergence [6]:(2)$$D\left(p,q\right)=H\left(\frac{p+q}{2}\right)-\frac{1}{2}H\left(p\right)-\frac{1}{2}H\left(q\right)$$
where *p* and *q* are the (relative) word frequencies of the two texts and *p* + *q* is calculated by concatenating both texts. From a Bayesian point of view, *D*(*p*,*q*) can be interpreted as the expected amount of gained information that comes from sampling one word token from the concatenation of both texts regarding the question which of the two texts the word token belongs to [7]. If the two texts are identical, *D*(*p*,*q*) = 0, because sampling a word token does not provide any information regarding to which text the token belongs. If, on the other side, the two texts do not have a single word type in common, then sampling one word token is enough to determine from which text the token comes, and correspondingly, *D*(*p*,*q*) = 1. The Jensen–Shannon divergence has already been applied in the context of measuring stylistic influences in the evolution of literature [8], cultural and institutional changes [9,10], the dynamics of lexical evolution [11,12], or to quantify changing corpus compositions [13]. 

Perhaps the most intriguing aspect of word frequency distributions is the fact that they can be described remarkably well by a simple relationship that is known as Zipf’s law [14]: if one assigns rank *r* = 1 to the most frequent word (type), rank *r* = 2 to the second most frequent word, and so on, then the frequency of a word and its rank *r* is related as follows:(3)$$p\left(r\right)\propto r^{-\gamma }$$
where the exponent *γ* is a parameter that has to be determined empirically. An estimation of *γ* by maximum likelihood (as described in [15]) for our database yields 1.10. However, when analyzing word frequency distributions, the main obstacle is that all quantities basically vary systematically with the sample size, i.e., the number of word tokens in the database [16,17]. To visualize this, we randomly arranged the order of all articles of our database. This step was repeated 10 times in order to create 10 different versions of our database. For each version, we estimate *H* and *γ* after every *n* = 2*^k^* consecutive tokens, where *k* = 6, 7, …, $log_{2}\left(N\right)=28$. Figure 1 shows a Simpson’s Paradox [18] for the resulting data: an apparent strong positive relationship between *H* and *γ* is observed across all datapoints (Spearman *ρ* = 0.99). However, when the sample size is kept constant, this relationship completely changes: if the correlation between *H* and γ is calculated for each *k*, the results indicate a strong negative relationship (*ρ* ranges between −0.98 and −0.64 with a median of −0.92). The reason for this apparent contradiction is the fact that both *H* and γ monotonically increase with the sample size. When studying word frequency distributions quantitatively, it is essential to take this dependence on the sample size into account [16].

Another important aspect of word distributions is the fact that word frequencies vary by a magnitude of many orders, as visualized in Figure 2. On the one hand, Figure 2a shows that there are very few word types that occur very often. For example, the 100 most frequent word types account for more than 40% of all word occurrences. Typically, many of those word types are function words [16] expressing grammatical relationships, such as adpositions or conjunctions. On the other hand, Figure 2b shows that there are a great deal of word types with a very low frequency of occurrence. For example, more than 60% of all word types only occur once, and less than 3% of all word types have a frequency of occurrence of more than 100 in our database. Many of those low frequency words are content words that carry the meaning of a sentence, e.g., nouns, (lexical) verbs, and adjectives. In addition to the sample size dependence outlined above, it is important to take this broad range of frequencies into account when quantitatively studying word frequency distributions [19].

In this context, it was recently demonstrated that generalized entropies of order *α*, also called Havrda–Charvat–Lindhard–Nielsen–Aczél–Daróczy–Tsallis entropies [20], offer novel and interesting opportunities to quantify the similarity of symbol sequences [21,22]. It can be written as:(4)$$H_{\alpha }\left(p\right)=\frac{1}{\alpha -1}\left(1-\overset{K}{\underset{i=1}{\sum }}p_{i}^{\alpha }\right)$$
where *α* is a free parameter. For α = 1, the standard Gibbs–Shannon entropy is recovered. Correspondingly, a generalization of the standard Jensen–Shannon divergence (Equation (2)) can be obtained by replacing *H* (Equation (1)) with *H*_α_ (Equation (4)) and thus leading to a spectrum of divergence measures *D*_α_, parametrized by α [22]. For the analysis of the statistical properties of natural languages, this parameter is highly interesting, because, as demonstrated by [21,22], varying the α-parameter allows us to magnify differences between different texts at specific scales of the corresponding word frequency spectrum. If α is increased (decreased), then the weight of the most frequent words is increased (decreased). As pointed out by an anonymous reviewer, a similar idea was already reported in the work of Tanaka-Ishii and Aihara [23], who studied a different formulation of generalized entropy, the so-called Rényi entropy of order α [24]. Because we are especially interested in using generalized entropies to quantify the (dis)similarity between two different texts or databases, following [21,22], we chose to focus on the generalization of Havrda–Charvat–Lindhard–Nielsen–Aczél–Daróczy–Tsallis instead of the formulation of Rényi, because a divergence measure based on the latter can become negative for α > 1 [25], while it can be shown that the corresponding divergence measure based on the former formulation is strictly non-negative [20,22]. In addition, *D*_α_(*p*,*q*) is the square of a metric for α∈(0,2], i.e., (i) *D*_α_(*p*,*q*) ≥ 0, (ii) *D*_α_(*p*,*q*) = 0 ⟺ *p* = *q*, (iii) *D*_α_(*p*,*q*) = *D*_α_(*q*,*p*), and (iv) $\sqrt{D_{\alpha }}$ obeys the triangular inequality [7,20,22].

In addition, [21] also estimated the size of the database that is needed to obtain reliable estimates of generalized divergences. For instance, [21] showed that only the 100 most frequent words contribute to *H*_α_ and *D*_α_ for α = 2.00, and all other words are practically irrelevant. This number quickly grows with α. For example, database sizes of *N* ≈ 10^8^ are needed for a robust estimation of the standard Jensen–Shannon divergence (Equation (2)), i.e., for α = 1.00. This connection makes the approach of [21,22] particularly interesting in relation to the systematic influence of the sample size demonstrated above (cf. Figure 1).

In this study, the approach is systematically and empirically studied by analyzing the lexical dynamics of the *Der Spiegel* periodical. The remainder of the paper is structured as follows: In the next section, details on the database and preprocessing are given (Section 2). In Section 3.1 and Section 3.2, the dependence of both *H*_α_ and *D*_α_ on the sample size is tested for different α-parameters. This section is followed by a case study, in which we demonstrate that the influence of sample size makes it difficult to quantify lexical dynamics and language change and also show that standard sampling approaches do not solve this problem (Section 3.3). This paper ends with some concluding remarks regarding the consequences of the results for the statistical analysis of languages (Section 4).

## 2. Materials and Methods

In the present study, we used all 365,514 articles that were published in the German weekly news magazine *Der Spiegel* between January 1947, when the magazine was first published, and December 2017. To read-in and tokenize the texts, we used the *Treetagger* with a German parameter file [26]. All characters were converted to lowercase. Punctuation and cardinal numbers (both treated as separate words by the Treetagger) were removed. However, from a linguistic point of view, changes in the usage frequencies of punctuation marks and cardinal numbers are also interesting. For instance, a frequency increase of the full stop could be indicative of decreases in syntactic complexity [15]. In Appendix B, we therefore present and discuss additional results in which punctuation and cardinal numbers were not removed from the data.

In total, our database consists of *N* = 236,743,042 word tokens and *K* = 4,009,318 different word types. 

Motivated by the studies of [21,22], we chose the following six α values to study the empirical behavior of generalized entropies and generalized divergences: 0.25, 0.75, 1.00, 1.50, and 2.00. To highlight that varying α makes it possible to magnify differences between different texts at specific scales of the corresponding word frequency spectrum, we take advantage of the fact that *H*_α_ can be written as a sum over different words, where each individual word type *i* contributes
(5)$$\begin{array}{c} \frac{p_{i}^{\alpha }-\frac{1}{K}}{\alpha -1},\;for\;\alpha \neq 1.00\\ -p_{i}*log_{2}\left(p_{i}\right),\;for\;\alpha =1.00 \end{array}.$$

In Table 1, we divided the word types into different groups according to their token frequency (column 1). Each group consists of *g* = 1, 2,…, *G* word types (cf. column 2). For each group, column 3 presents three randomly chosen examples.

This implies that the relative contribution *C*(*g*) per group can be calculated as (see also ([21], Equation (5))):(6)$$C\left(g\right)=\{\begin{array}{c} \frac{\sum_{g=1}^{G}p_{g}^{\alpha }}{\sum_{i=1}^{K}p_{i}^{\alpha }},\;for\;\alpha \neq 1.00\\ \frac{\sum_{g=1}^{G}\left(-1\right)*p_{g}*log_{2}(p_{g)}}{\sum_{i=1}^{K}\left(-1\right)*p_{i}*log_{2}(p_{i)}},\;for\;\alpha =1.00 \end{array}.$$

Columns 4–8 of Table 1 show the relative contribution (in %) for each group to *H*_α_ as a function of α. For lower values of α, *H*_α_ is dominated by word types with lower token frequencies. For instance, hapax legomena, i.e., word types that only occur once, contribute almost half of *H*_α=0.25_. For larger values of α, only the most frequent word contributes to *H*_α_. For example, the 27 word types with a token frequency of more than 1,000,000 contribute more than 92% to *H*_α=2.00_. Because words in different frequency ranges have different grammatical and pragmatic properties, varying α makes it possible to study different aspects of the word frequency spectrum [21].

As written above, we are interested in testing the dependence of both *H*_α_ and *D*_α_ on the sample size for the different *α*-values. Let us note that each article in our database can be described by different attributes, e.g., publication date, subject matter, length, category, or author. Of course, this list of attributes is not exhaustive but can be freely extended depending on the research objective. In order to balance the article’s characteristics across the corpus, we prepared 10 versions of our database, each with a different random arrangement of the order of all articles. To study the convergence of *H*_α_, we computed *H*_α_ after every *n* = 2*^k^* consecutive tokens for each version, where *k* = 6, 7, …, $log_{2}\left(N\right)=27$. For *D*_α_, we compared the first *n* = 2*^k^* word tokens with the last *n* = 2*^k^* of each version of our database. Here, *k* = 6, 7, …, 26. For instance for *k* = 26, the first 67,108,864 word tokens are compared with the last 67,108,864 word tokens by calculating the generalized divergence between both “texts” for different *α*-values. Through the manipulation of the article order, it can be inferred that, random fluctuations aside, any systematic differences are caused by differences in the sample size.

As outlined above, our initial research interest concerned the use of generalized entropies and divergence in order to measure lexical change rates at specific ranges of the word frequency spectrum. To this end, we used the publication date of each article on a monthly basis to create a diachronic version of our database. Figure 3 visualizes the corpus size *N_t_* for each *t*, where each monthly observation is identified by a variable containing the year *y* = 1947, 1948, …, 2017 and the month *m* = 1, 2, …,12.

Instead of calculating the generalized Jensen–Shannon divergences for two different texts *p* and *q*, *D*_α_ was calculated for successive moments in time, i.e., *D_α_*(*t*,*t* − 1), in order to estimate the rate of lexical change at a given time point *t* [11,12]. For instance, *D*_α_ at *y* = 2000 and *m* = 1 represents the generalized divergence for a corresponding *α*-value between all articles that were published in January 2000 and those published in December 1999. The resulting series of month-to-month changes could then be analyzed in a standard time-series analysis framework. For example, we can test whether the series exhibits any large-scale tendency to change over time. A series with a positive trend increases over time, which would be indicative of an increasing rate of lexical change. It would also be interesting to look at first differences in the series, as an upward trend here in addition to an upward trend in the actual series would mean that the rate of lexical change is increasing at an increasing rate. 

However, because the sample size clearly varies as a function of time (cf. Figure 3), it was essential to rule out the possibility that this variation systematically influences the results. Therefore, we generated a second version of this diachronic database in which we first randomly arranged the order of each article again. We then used the first *N*_t=1_ words of this version of the database to generate a new corpus that has the same length (in words) as the original corpus at *t* = 1 but in which the diachronic signal is destroyed. We then proceeded and used the next *N*_t=2_ words to generate a corpus that has the same length as the original corpus at *t* = 2. For example, the length of a concatenation of all articles that where published in *Der Spiegel* in January 1947 is 94,716 word tokens. Correspondingly, our comparison corpus at this point in time also consisted of 94,716 word tokens, but the articles of which it consisted could belong to any point in time between 1947 and 2017. In what follows, we computed all *D_α_* (*t*,*t* − 1) values for both the original version of our database and for the version with a destroyed diachronic signal. We tentatively called this a “Litmus test”, because it determined whether our results can be attributed to real diachronic changes or if there is a systematic bias due to the varying sample sizes.

*Statistical analysis*: To test if *H*_α_ and *D*_α_ vary as a function of the sample size without making any assumptions regarding the functional form of the relationship, we used the non-parametric Spearman correlation coefficient denoted as *ρ*. It assesses whether there is a monotonic relationship between two variables and is computed as Pearson’s correlation coefficient on the ranks and average ranks of the two variables. The significance of the observed coefficient was determined by Monte Carlo permutation tests in which the observed values of the sample size are randomly permuted 10,000 times. The null hypothesis is that *H*_α_/*D*_α_ does not vary with the sample size. If this is the case, then the sample size becomes arbitrary and can thus be randomly re-arranged, i.e., permuted. Let *c* denote the number of times the absolute *ρ*-value of the derived dataset is *greater than or equal to* the absolute *ρ*-value computed on the original data. A corresponding coefficient was labeled as “statistically significant” if *c* < 10, i.e., *p* < 0.001. In cases where *l*, i.e., the number of datapoints, was lower than or equal to 7, an exact test for all *l*! permutations was calculated. Here, let *c** denote the number of times where the absolute *ρ*-value of the derived dataset is *greater than* the absolute *ρ*-value computed on the original data. A coefficient was labeled as “statistically significant” if *c**/*l*! < 0.001. 

*Data availability and reproducibility*: All datasets used in this study are available in Appendix A (https://doi.org/10.7910/DVN/OP9PRL). For copyright and license reasons, each actual word type is replaced by a unique numerical identifier. Regarding further data access options, please contact the corpus linguistics department at Institute for the German language (IDS) (korpuslinguistik@ids-mannheim.de). In the spirit of reproducible science, one of the authors (A.K.) first analyzed the data using Stata and prepared a draft. Another author (S.W.) then used the draft and the available datasets to reproduce all the results using R. The results of this replication are available and the code (Stata and R) required to reproduce all the results presented in this paper are available in Appendix A (https://doi.org/10.7910/DVN/OP9PRL).

## 3. Results

### 3.1. Entropy H_α_

To test the sample size dependence of *H_α_*, we computed *H_α_* for the first *n* = 2*^k^* consecutive tokens, where *k* = 6, 7, …, 27 for the 10 versions of our database (each with a different random article order) and calculated averages. Figure 4A shows the convergence pattern for the five *α*-values in a superimposed scatter plot with connected dots where the colors of each *y*-axis correspond to one *α*-value (cf. the legend in Figure 4, the axes are log-scaled for improved visibility). For values of *α* < 1.00, there is no indication of convergence, while for *H_α=1.50_* and *H_α=2.00_*, it seems that *H_α_* converges rather quickly. To test the observed relationship between the sample size and *H_α_* for different *α*-values, we calculated the Spearman correlation between the sample size and *H_α_* for different minimum sample sizes. For example, a minimum sample size of *n* = 2^17^ indicates that we restrict the calculation to sample sizes ranging between *n* = 2^17^ and *n* = 2^27^. For those 11 datapoints, we computed the Spearman correlation between the sample size and *H_α_* and ran the permutation test. Table 2 summarizes the results. For all α-values, except for *α* = 2.00, there is a clear indication for a significant (at *p* < 0.001) strong, positive, monotonic relationship between *H_α_* and the sample size for all the minimum sample sizes. Thus, while Figure 4A seems to indicate that *H_α=1.50_* converges rather quickly, the Spearman analysis reveals that the sample size dependence of *H_α=1.50_* persists for higher values of *k* with a minimum *ρ* of 0.80. Except for the last two minimum sample sizes, all the coefficients pass the permutation test. For *α* = 2.00, *H_α_* starts to converge after *n* = 2^14^ word tokens. None of the correlation coefficients of higher minimum sample sizes passes the permutation test. In line with the results of [21,22], this suggest *α* = 2.00 as a pragmatic choice when calculating *H_α_*. However, it is important to point out that for *α* = 2.00, the computation of *H_α_* is almost completely determined by the most frequent words (cf. Table 1). For lower values of α, the basic problem of sample size dependence (cf. Figure 1) persists. If it is the aim of a study to compare *H_α_* for databases with varying sizes, this has to be taken into account. Correspondingly, [23] reached similar conclusions for the convergence of Rényi entropy of order α = 2.00 for different languages and for different kinds of texts, both on the level of words and on the level of characters. In Appendix C, we have replicated the results of Table 2 based on Rényi’s formulation of the entropy generalization. Table A5 shows that the results are almost identical, which is to be expected because the Havrda–Charvat–Lindhard–Nielsen–Aczél–Daróczy–Tsallis entropy is a monotone function of the Rényi entropy [20].

### 3.2. Divergence D_α_

To test the relationship between the sample size and *D_α_* for different α-values, we computed *D_α_* for a “text” that consists of the first *n* = 2*^k^* word tokens, a “text” that consists of the last *n* = 2*^k^* word tokens for each version of our database for *k* = 6, 7, …, 26, and took averages. As for *H_α_* above, we then calculated the Spearman correlation between the sample size and *D*_α_ for different minimum sample sizes. It is worth pointing out that the idea here is that the “texts” come from the same population, i.e., all *Der Spiegel* articles, so one should expect that with growing sample sizes, *D*_α_ should fluctuate around 0 with no systematic relationship between *D*_α_ and the sample size. Table 3 summarizes the results, while Figure 4B visualizes the convergence pattern. For all settings, there is a strong monotonic relationship between the sample size and *D_α_* that passes the permutation test in almost every case. For *α* = 0.25, the Spearman correlation coefficients are positive. This seems to be due to the fact that *H_α=0.25_* is dominated by word types from the lower end of the frequency spectrum (cf. Table 1). Because, for example, word types that only occur once contribute almost half of *H*_α=0.25_. Those word types then either appear in the first 2*^k^* or in the last 2*^k^* word tokens.

The results demonstrate that the larger the sample sizes the larger *D_α_* (cf. the pink line in Figure 4B). For = 0.75, a similar pattern is observed for smaller sample sizes (cf. the orange line in Figure 4 B). However, at around *k* = 15, the pattern changes. For *k* ≥ 15, there is a perfect monotonic negative relationship between *D_α=0.75_* and the sample size. Surprisingly, there is a perfect monotonic negative relationship for all settings for *α* ≥ 1.00, even if we restrict the calculation to relatively large sample sizes. However, the corresponding values are very small. For instance, *D_α=2.00_* = 7.91 × 10^−8^ for *n* = 2^24^, *D_α=2.00_* = 4.08 × 10^−8^ for *n* = 2^25^, and *D_α=2.00_* = 1.379 × 10^−8^ for *n* = 2^26^. One might object that this systematic sample size dependence is practically irrelevant. In the next section, we show that, unfortunately, this is not the case.

### 3.3. Case Study

As previously outlined, our initial idea was to use generalized divergences to measure the rate of lexical change at specific ranges of the word frequency spectrum. In what follows, we estimate the rate by calculating *D_α_* for successive months, i.e., *D_α_*(*t*,*t* − 1). To rule out a potential systematical influence of the varying sample size, we also calculated *D_α_*(*t*,*t* − 1) for our comparison corpus where the diachronic signal was destroyed (“Litmus test”).

For *α*, we chose 2.00 and 1.00. On the one hand, the analyses of [21,22] and our analysis presented above indicate that *α* = 2.00 seems to be the most robust choice. On the other hand, we chose *α* = 1.00, i.e., the original Jensen–Shannon divergence, because, as explained above, it has already been employed in the context of analyzing natural language data without explicitly testing the potential influence of varying sample sizes. Figure 5 shows our results. If we only looked at the plots on the left side (blue lines), the results would look very interesting, as there is a clear indication that the rate of lexical change decreases as a function of time for both *α* = 1.00 and for *α* = 2.00. However, looking at the plots in the middle reveals that a very similar pattern emerges for the comparison data. For our “Litmus test”, we destroyed all diachronic information except for the varying sample sizes. Nevertheless, our conclusions would have been more or less identical. Interestingly, the patterns in Figure 5 clearly resemble the pattern of the sample size in Figure 3 (in reverse order) and thus suggest a negative association between *D_α_*(*t*,*t* − 1) and the sample size. To test this observation, we calculated the Spearman correlation between the sample size and *D_α_*(*t*,*t* − 1) for both *α* = 1.00 and *α* = 2.00 and ran a permutation test. Table 4, row 1, shows that there is a significant strong negative correlation between the sample size and *D_α_* for both *α* = 1.00 and *α* = 2.00. Rows 2–5 present different approaches to solving the sample size dependence of *D_α_*. In row 2, we extended Equation (2) to allow for unequal sample sizes, i.e., *N_p_* ≠ *N_q_* as suggested by ([22], Appendix B); here: (7)$$\begin{array}{c} D_{\alpha }^{\pi }\left(p,q\right)=H_{\alpha }\left(\pi_{p}p+\pi_{q}q\right)-\pi_{p}H_{\alpha }\left(p\right)-\pi_{q}H_{\alpha }\left(q\right)\\ \mathrm{where}\;\pi_{p}=N_{p}/\left(N_{p}+N_{q}\right)\;\mathrm{and}\;\pi_{q}=N_{q}/\left(N_{p}+N_{q}\right). \end{array}$$

Row 2 of Table 4 demonstrates that this “natural weights” approach does not qualitatively affect the results; there is still a significant and strong negative correlation between the sample size and $D_{\alpha }^{\pi }$ for both *α* = 1.00 and *α* = 2.00. Another approach is to increase the sample size (if possible). To this end, we aggregated the articles at the annual level instead of the monthly level. On average, the annual corpora are $\bar{N}$ = 3,334,409.04 words long, compared to $\bar{N}$ = 277,867.42 word tokens for the monthly data. Row 3 of Table 4 shows that increasing the sample size does not help in removing the influence of the sample size either. Another standard approach [15,22] is to randomly draw *N_min_* word tokens from the monthly databases, where *N_min_* is equal to the smallest of all monthly corpora, here *N_min_* = 75,819 (June 1947). To our own surprise, row 4 of Table 4 reveals that this “random draw” approach also does not break the sample size dependence. While the absolute values of the correlation coefficients for both *α* = 1.00 and *α* = 2.00 are smaller for the original data than for the comparison data, all four coefficients are significantly different from 0 (at *p* < 0.001) and thus indicate that the “random draw” approach fails to pass the “Litmus test”. As a last idea, we decided to truncate each monthly corpus after *N_min_* word tokens. The difference between this “cut-off” approach and the “random draw” is that the latter approach assumes that words occur randomly in texts, while truncating the data after *N_min_* as in the “cut-off” approach respects the syntactical and semantical coherence and the discourse structure at the text level [16,17]. On the one hand, row 5 of Table 4 demonstrates that this approach mostly solves the problem: all four coefficients are small, and only one coefficient is significantly different from zero, but positive. This suggests that the “cut-off” approach passes the “Litmus test”. On the other hand, it’s worth pointing out that we lose a lot of information with this approach. For example, the largest corpus is *N* = 507,542 word tokens long (October 2000). With the “cut-off” approach, more than 85% of those word tokens are not used to calculate *D_α_*(*t*,*t* − 1).

While the resulting pattern in Figure 6 might be indicative of an interesting lexico-dynamical process, especially for α = 1.00, what is more important in the present context is the fact that both blue lines look completely different compared with the corresponding blue lines in Figure 5. Thus, in relation to the analysis above (cf. Section 3.2), we concluded that the systematic sample size dependence of *D_α_* is far from practically irrelevant. On the contrary, the analyses presented in this section demonstrate again why it is essential to account for the sample size dependence of lexical statistics.

## 4. Discussion

In this paper, we explored the possibilities of using generalized entropies to analyze the lexical dynamics of natural language data. Using the α-parameter in order to automatically magnify differences between different texts at specific scales of the corresponding word frequency spectrum is interesting, as it promises a more objective selection method compared to, e.g., [8], who use a pre-compiled list of content-free words, or [12], who analyzes differences within different part-of-speech classes.

In line with other studies [17,23,27,28,29], the results demonstrate that it is essential for the analysis of natural language to always take into account the systematic influence of the sample size. With the exception of *H_α=2.00_* for larger sample sizes, all quantities that are based on general entropies seem to strongly covary with the sample size (also see [23] for similar results based on Rényi’s formulation of generalized entropies). In his monograph on word frequency distributions, Baayen [16] introduces the two fundamental methodological issues in lexical statistics:
The sample size crucially determines a great many measures that have been proposed as characteristic text constants. However, the values of these measures change systematically as a function of the sample size. Similarly, the parameters of many models for word frequency distribution [sic!] are highly dependent on the sample size. This property sets lexical statistics apart from most other areas in statistics, where an increase in the sample size leads to enhanced accuracy and not to systematic changes in basic measures and parameters.… The second issue concerns the theoretical assumption […] that words occur randomly in texts. This assumption is an obvious simplification that, however, offers the possibility of deriving useful formulae for text characteristics. The crucial question, however, is to what extent this simplifying assumption affects the reliability of the formulae when applied to actual texts and corpora.(p.1)

The main message of this paper is that those two fundamental issues also pose a strong challenge to the application of information theory for the quantitative study of natural language signals. In addition, the results of the case study (cf. Section 3.3) indicate that both fundamental issues in lexical statistics apparently interact with each other. As mentioned above, there are numerous studies that used the Jensen–Shannon divergence or related measures without an explicit “Litmus test”. Let us mention two examples from our own research:(i)In [12], an exploratory data-driven method was presented that extracts word-types from diachronic corpora that have undergone the most pronounced change in frequency of occurrence in a given period of time. To this end, a measure that is approximately equivalent to the Jensen–Shannon divergence is computed and period-to-period changes are calculated as in Section 3.3.(ii)In [15], the parameters of the Zipf–Mandelbrot law were used to quantify and visualize diachronic lexical, syntactical, and stylistic changes, as well as aspects of linguistic change for different languages.

Both studies are based on data from the Google Books Ngram corpora, made available by [30]. It contains yearly token frequencies for each word type for over 8 million books, i.e., 6% of all books ever published [31]. To avoid a potential systematic bias due to strongly changing corpus sizes, random samples of equal size were drawn from the data in both [12] and [15]. However, as demonstrated in Section 3.3, apparently this simplifying assumption is problematic, because it seems to make a difference if we randomly sample *N* word tokens or if we keep the first *N* word tokens for the statistical structure of the corresponding word frequency distribution. It is worth pointing out again that, without the “Litmus test” the interpretation of the results presented in Section 3.3 would have been completely different, because randomly drawing word tokens from the data does not seem to break the sample size dependence. It is an empirical question whether the results presented in [12,15], and comparable other papers would pass a “Litmus test”. In light of the results presented in this paper, we are rather skeptical, thus echoing the call of [22] that it is “essential to clarify what is the role of finite-size effects in the reported conclusions, in particular in the (typical) case that database sizes change over time.” (p. 8). One could even go so far as to ask whether relative frequencies that are compared between databases of different sizes are systematically affected by varying database sizes. However, the test scheme as we introduced it presupposes access to the full text data. For instance, due to copyright constraints, access to Google Books Ngram data is restricted to token frequencies for all words (and phrases) that occur at least 40 times in the corpus. Thus, an analogous “Litmus test” is not possible. At our institute, we are rather fortunate to have access to the full text data of our database. Notwithstanding, copyright and license reasons are a major issue here, as well [32]. To solve this problem for our study, we replaced each actual word type with a unique numerical identifier as explained in Section 3.3. For our focus of research, using such a pseudonymization strategy is fine. However, there are many scenarios where, depending on the research objective, the actual word strings matter, making it necessary to develop a different access and publication strategy. It goes without saying that, in all cases, full-text access is the best option.

While the peculiarities of word frequency distributions make the analysis of natural language data more difficult compared to other empirical phenomena, we hope that our analyses (especially the “Litmus test”) also demonstrate that textual data offer novel possibilities to answer research questions. Or put differently, natural language data contain a lot of information that can be harnessed. For example, two reviewers pointed out that it could make sense to develop a method that recovers an unbiased lexico-dynamical signal by removing the “Litmus test” signal from the original signal. This is an interesting avenue for future research.

## Figures and Tables

**Figure 1 entropy-21-00464-f001:**
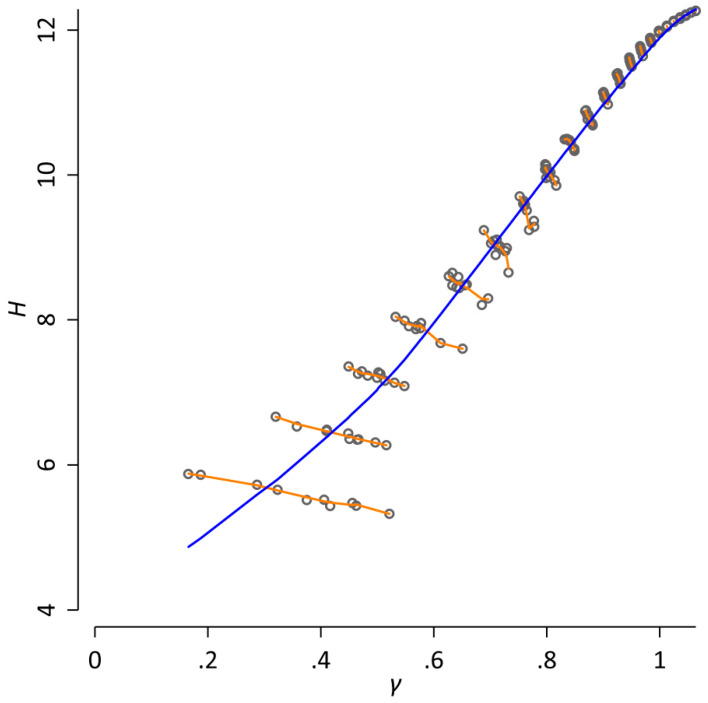
A Simpson’s Paradox for word frequency distributions. Here, the word entropy *H* and the exponent of the Zipf distribution *γ* are estimated after every *n* = 2*^k^* consecutive tokens, where *k* = 6, 7, …, $log_{2}\left(N\right)$ for 10 different random re-arrangements of the database; each dot corresponds to one observed value. The blue line represents a locally weighted regression of *H* on *γ* (with a bandwidth of 0.8). It indicates a strong positive relationship between *H* and *γ* (Spearman *ρ* = 0.99). However, when the sample size is held constant, this relationship completely changes, as indicated by the orange lines that correspond to separate locally weighted regressions of *H* on γ for each *k*. Here, the results indicate a strong negative relationship between H and *γ* (*ρ* ranges between −0.98 and −0.64 with a median of −0.92). The reason for this apparent contradiction is the fact that both H and *γ* monotonically increase with the sample size.

**Figure 2 entropy-21-00464-f002:**
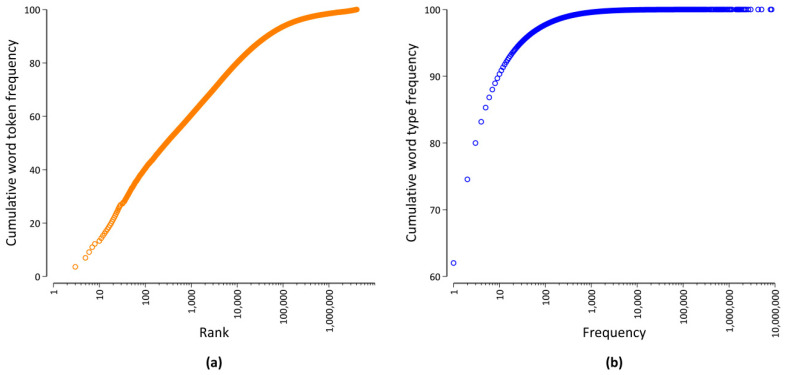
Visualization of the word frequency distribution of our database. Cumulative distribution (in %) as a function of (**a**) the rank and (**b**) the word frequency.

**Figure 3 entropy-21-00464-f003:**
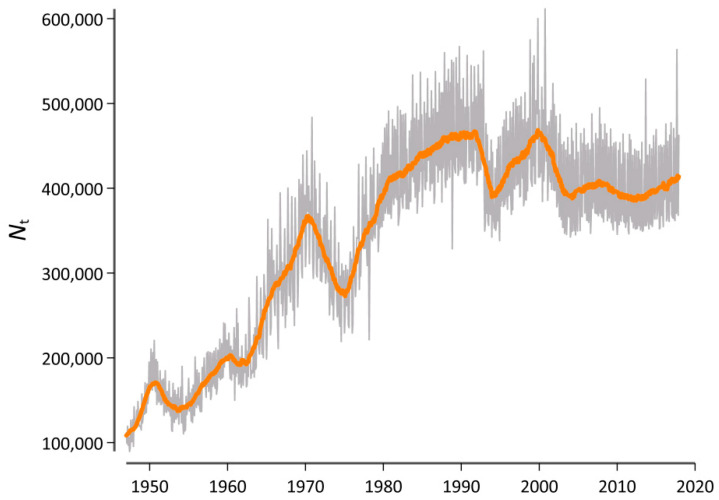
Sample size of the database as a function of time. The gray line depicts the raw data, while the orange line adds a symmetric 25-month window moving-average smoother highlighting the central tendency of the series at each point in time.

**Figure 4 entropy-21-00464-f004:**
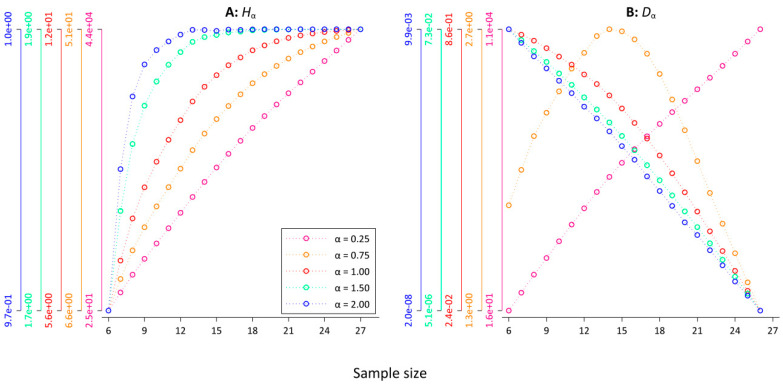
Generalized entropies *H_α_* and divergences *D_α_* as a function of the sample size. (**A**) *P*_α,_ (**B**) *D_α_*.

**Figure 5 entropy-21-00464-f005:**
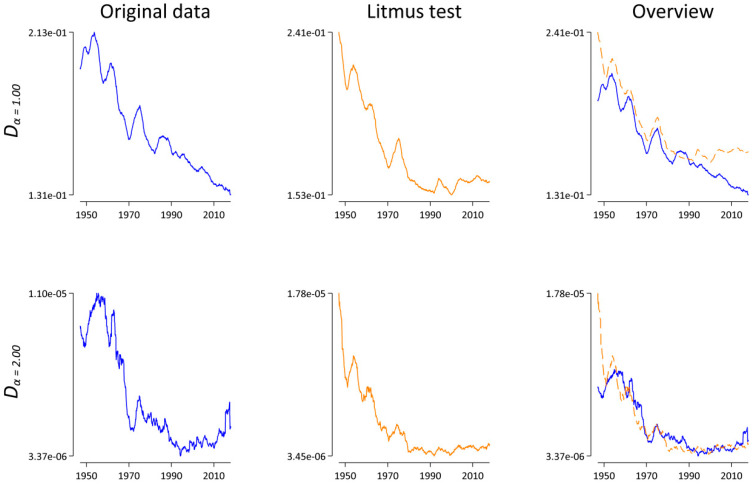
*D_α_*(*t*,*t* − 1) as a function of time for *α* = 1.00 and *α* = 2.00. Lines represent a symmetric 25-month window moving-average smoother highlighting the central tendency of the series at each point in time. Left: results for the original data in blue. Middle: results for the “Litmus” data in orange. Right: superimposition of both the original and the “Litmus” data.

**Figure 6 entropy-21-00464-f006:**
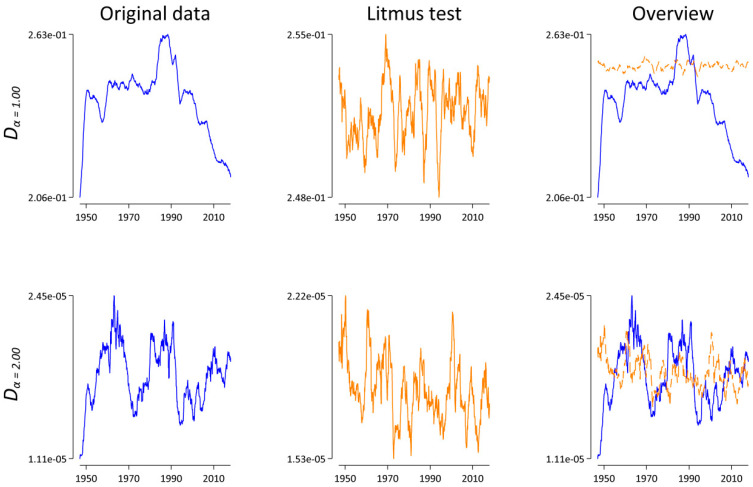
*D_α_*(*t*,*t* − 1) as a function of time for *α* = 1.00 and *α* = 2.00. Here, each monthly corpus is truncated after *N_min_* = 75,819 word tokens. Lines represent a symmetric 25-month window moving-average smoother highlighting the central tendency of the series at each point in time. Left: results for the original data in blue. Middle: results for the “Litmus” data in orange. Right: superimposition of both the original and the “Litmus” data.

**Table 1 entropy-21-00464-t001:** Contribution (in %) of word types with different token frequencies as a function of α *.

Token Frequency	Number of Cases	Examples	α = 0.25	α = 0.75	α = 1.00	α = 1.50	α = 2.00
1	2,486,393	koalitionsbündnisse nr.6/1962 bruckner-breitklang	48.65	9.32	2.38	0.00	0.00
2–10	1,135,102	geschlechterschulung unal wiedervereinigungs-prozedur	29.86	10.89	3.65	0.01	0.00
11–100	296,573	hotpants lánský planwirtschaftlichen	13.16	14.03	7.13	0.04	0.00
101–1000	74,791	wanda verbannte mitschnitt	5.83	19.21	14.69	0.28	0.00
1001–10,000	14,388	schüren ablesen vollmachten	1.96	19.81	22.07	1.53	0.06
10,001–100,000	1871	london sitzen beginnen	0.44	13.38	20.68	5.31	0.64
100,001–1,000,000	173	mark frau kaum	0.07	7.38	15.21	17.83	7.12
1,000,001 +	27	es die er	0.02	5.98	14.19	75.02	92.18
	4,009,318		100.00	100.00	100.00	100.00	100.00

* Values are rounded for illustration purposes only throughout this paper.

**Table 2 entropy-21-00464-t002:** Spearman correlation between the sample size and *H_α_* for different α-values *.

Minimum Sample Size	Number of Datapoints	α = 0.25	α = 0.75	α = 1.00	α = 1.50	α = 2.00
2^6^	22	1.00 *	1.00 *	1.00 *	1.00 *	0.92 *
2^7^	21	1.00 *	1.00 *	1.00 *	1.00 *	0.91 *
2^8^	20	1.00 *	1.00 *	1.00 *	1.00 *	0.89 *
2^9^	19	1.00 *	1.00 *	1.00 *	1.00 *	0.87 *
2^10^	18	1.00 *	1.00 *	1.00 *	1.00 *	0.85 *
2^11^	17	1.00 *	1.00 *	1.00 *	1.00 *	0.82 *
2^12^	16	1.00 *	1.00 *	1.00 *	1.00 *	0.79 *
2^13^	15	1.00 *	1.00 *	1.00 *	1.00 *	0.74
2^14^	14	1.00 *	1.00 *	1.00 *	1.00 *	0.71
2^15^	13	1.00 *	1.00 *	1.00 *	0.99 *	0.65
2^16^	12	1.00 *	1.00 *	1.00 *	0.99 *	0.55
2^17^	11	1.00 *	1.00 *	1.00 *	0.99 *	0.43
2^18^	10	1.00 *	1.00 *	1.00 *	0.99 *	0.24
2^19^	9	1.00 *	1.00 *	1.00 *	0.98 *	−0.05
2^20^	8	1.00 *	1.00 *	1.00 *	0.98 *	−0.17
2^21^	7	1.00 *	1.00 *	1.00 *	0.96 *	0.25
2^22^	6	1.00 *	1.00 *	1.00 *	0.94	−0.20
2^23^	5	1.00 *	1.00 *	1.00 *	0.90	0.10
2^24^	4	1.00 *	1.00 *	1.00 *	0.80	−0.80

* An asterisk indicates that the corresponding correlation coefficient passed the permutation test at *p* < 0.001. For minimum sample sizes above 2^20^, an exact permutation test is calculated.

**Table 3 entropy-21-00464-t003:** Spearman correlation between the sample size and *D_α_* for different α-values *.

Minimum Sample Size	Number of Datapoints	α = 0.25	α = 0.75	α = 1.00	α = 1.50	α = 2.00
2^6^	21	1.00 *	−0.42	−1.00 *	−1.00 *	−1.00 *
2^7^	20	1.00 *	−0.54	−1.00 *	−1.00 *	−1.00 *
2^8^	19	1.00 *	−0.64	−1.00 *	−1.00 *	−1.00 *
2^9^	18	1.00 *	−0.74	−1.00 *	−1.00 *	−1.00 *
2^10^	17	1.00 *	−0.83 *	−1.00 *	−1.00 *	−1.00 *
2^11^	16	1.00 *	−0.90 *	−1.00 *	−1.00 *	−1.00 *
2^12^	15	1.00 *	−0.95 *	−1.00 *	−1.00 *	−1.00 *
2^13^	14	1.00 *	−0.99 *	−1.00 *	−1.00 *	−1.00 *
2^14^	13	1.00 *	−1.00 *	−1.00 *	−1.00 *	−1.00 *
2^15^	12	1.00 *	−1.00 *	−1.00 *	−1.00 *	−1.00 *
2^16^	11	1.00 *	−1.00 *	−1.00 *	−1.00 *	−1.00 *
2^17^	10	1.00 *	−1.00 *	−1.00 *	−1.00 *	−1.00 *
2^18^	9	1.00 *	−1.00 *	−1.00 *	−1.00 *	−1.00 *
2^19^	8	1.00 *	−1.00 *	−1.00 *	−1.00 *	−1.00 *
2^20^	7	1.00 *	−1.00 *	−1.00 *	−1.00 *	−1.00 *
2^21^	6	1.00 *	−1.00 *	−1.00 *	−1.00 *	−1.00 *
2^22^	5	1.00 *	−1.00 *	−1.00 *	−1.00 *	−1.00 *
2^23^	4	1.00 *	−1.00 *	−1.00 *	−1.00 *	−1.00 *
2^24^	3	1.00 *	−1.00 *	−1.00 *	−1.00 *	−1.00 *

* An asterisk indicates that the corresponding correlation coefficient passed the permutation test at *p* < 0.001. For minimum sample sizes above 2^19^, an exact permutation test is calculated.

**Table 4 entropy-21-00464-t004:** Spearman correlation between the sample size and *D_α_*(*t*,*t* − 1) for the original data and for the “Litmus test” for *α* = 1.00 and *α* = 2.00.

Row	Scenario	α	Number of Cases	Original Data	Litmus Test
1	Original	1.00	851	−0.76 *	−0.91 *
		2.00	851	−0.70 *	−0.79 *
2	Natural weights	1.00	851	−0.77 *	−0.90 *
		2.00	851	−0.70 *	−0.79 *
3	Yearly data	1.00	70	−0.74 *	−0.97 *
		2.00	70	−0.46 *	−0.87 *
4	Random draw	1.00	851	−0.16 *	−0.69 *
		2.00	851	−0.50 *	−0.61 *
5	Cut-off	1.00	851	0.12 *	0.08
		2.00	851	0.08	−0.10

* An asterisk indicates that the corresponding correlation coefficient passed the permutation test at *p* < 0.001.

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
