# Peer review of "Studying Lexical Dynamics and Language Change via Generalized Entropies: The Problem of Sample Size"

_entropy, 2019, doi:10.3390/e21050464_

Round 1
Reviewer 1 Report
In this manuscript the authors investigate the use of information-theoretic measures (those based on the generalized entropies of order alpha) in studying language change over time. Using a large dataset of news articles from the past 70 years, they perform a range of experiments showing empirically how differences in sample size affect the measurements; and thus the conclusions we draw about the rate of language change over time.
Information-theoretic measures (such as the Jensen-Shannon divergence) have become extremely popular in applications of computational linguistics in the past years. They not only offer a theoretically well-founded approach to measure language change but also provide an elegant solution to take into account the strong variability of word frequencies due to Zipf’s law. However, one of the most crucial but typically ignored (and thus unsolved) issues is the dependence of these measures on sample size. The current manuscript provides much needed new insights into this question in order to ensure the reliability of the conclusions drawn from such measurements. In this, I find that the work aligns strongly with the scope of Entropy. The statistical analysis is extensive and the manuscript is well-written.
Therefore I recommend publication of the manuscript in Entropy in principle – I do have 2 main concerns about the statistical analysis, which hope the authors can address and believe would strengthen the claims of the manuscript.
1. I liked very much that the authors compared different techniques that have been proposed to account for the sample size issue in practice, showing that the derived lexico-dynamical processes shown in Fig. 5 and 6 look quite differently. In this sense the results (and the discussion) seem rather negative. However, when looking at the plots, it seems to me that there is a way to reconcile the two seemingly results by interpreting the “Litmus-test”-curve as a null model: The “true” lexico-dynamical signal is thus only the difference between the measured curve (left) and the null model curve (middle), e.g. in form of a z-score. For example, for alpha=1, both Fig. 5 and 6 show an excess of the blue curve over the orange curve around the year 1990. Visually, it would seem that the same could hold true for alpha = 2 (i.e. that the difference between blue and orange is in fact very similar).
More generally, this could hint that the aim of a “Litmus-test” which should be “passed” is too strict of a criterion. Instead, one might be able to account for the sample size effect by simply “subtracting” the contribution from data in which the signal which one intends to measure was removed. Here, the “Litmus-test” data proposed by the authors is an elegant way to do that for the lexico-dynamical signal.
2. The convergence analysis of H_alpha and D_alpha in Sec. 3.1 and 3.2 shows how difficult it is to accurately measure the information-theoretic quantities even in large datasets. While I understand Fig. 4, I am not convinced that the correlation coefficients in Table 2 and 3 are really meaningful. In the case studied here, the measured value of D_alpha will always be correlated with N, no matter how small the bias is (we know the bias becomes infinitely small). The only possibility we would not measure a correlation is if i) the actual signal is larger than the bias (which in this case does not apply since the two samples compared come from the randomized data); or ii) the statistical error (the variance) is larger than the bias. However, my hunch is that the variance will be much smaller than the bias in this case. Therefore, the correlation analysis ignores the effect size of the sample size dependence.
Minor points:
- line 44: The sentence “In the present paper, ...” seemed to break the explanation on H(p) in the paragraph. Perhaps it could easily merge with the sentence in line 137 at the end of the same section “In this study, ...”
- Figure 2: From what I understand, the y-axis label refers to percentage of word tokens on the left, while on the right it is percentage of word types? Perhaps it would be easier to parse if this information is incorporated into the label or the caption.
- Figure 4,5,6: I think errorbars (over the 10 different realizations of the random ordering) are crucial to interpret the results.
- References 17 and 25 are duplicates.
p { margin-bottom: 0.25cm; line-height: 120%; }Author Response
please see the attached pdf file

Reviewer 2 Report
The authors must consider the following article:
Kumiko Tanaka-Ishii and Shunsuke Aihara. Computational Constancy Measures of Texts-Yule’s K and Rényi’s Entropy. Computational Linguistics , 41(3): 481--502, 2015.
This work, as far as I understand, considered the measures that do not rely on sample size, and raises Renyi generalized entropy for a larger alpha, as one such measure. It shows the speed of convergence of the various measures w.r.t. sample size.
I rather find Table 2 and 3 are obvious, given their results.
The above work suggest how large alpha requires less sample amount to acquire a stable result with respect to the entropy.
It is good that the conclusion of this submission somewhat agrees with their report. I think that the article can be much improved by reorganizing the
conclusion based on the above journal paper.
Further, the authors used Tsallis generalized entropy based on the previous work, there are other generalized entropy as Renyi, as considered in the above article. The one reason why the convergence is slow with Tsallis is that Tsallis does not take the log of the probability sum, whereas Renyi entropy does. Therefore, by use of Renyi generalized entropy, the conclusion should change. I suggest that authors reorganize a discussion or preferably by adding an experimental support using Renyi entropy.
--------
You should note how you fitted gamma. In Figure 1, gamma seems to vary
greatly with respect to sample size. Is it relevant to fit a text of so small
size of k=6,7 to gamma?

Author Response
please see the attached pdf file.

Reviewer 3 Report
Studying lexical dynamics and language change via generalized entropies – the problem of sample size
Review
This study analyzes the impact of the size of the text-sample on the Entropy and Divergence Entropy computed for the text. To set their experiment, the authors use the Generalized Entropy expression instead of the conventional Shannon’s entropy. This extends the scope of their study. The conclusions indicate a relevant influence over text length over computed entropy and divergence
Some comments on the manuscript writing.
The writing in this manuscript is clear and easy to read. This allows for a fluent reading leading to the understanding of the actual paper’s content.
In the Abstract the use of the symbol alpha looks premature. It is not easy to understand what this symbol represents. The Abstract is almost entirely referred to this parameter alpha. Thus, it seems appropriate to write the meaning of alpha the first time it is mentioned.
Line 17: Consider ‘systematically and empirically’ instead of ‘systematically empirically’:
Comments
Line 46: Does German actually have more than 4 million words? It looks hi since English does not reach 1 million. Do the authors think this may be due to the way German combines words to form compound tokens which in turn can be considered words? If so, this represents an important structural difference between German and the writing system of other natural languages.
Page 2. Lines 63-67. While explaining the meaning of D(p, q) values, the authors refer to word tokens in texts p and q and explain the value of D(p, q) in terms of the frequency some words may appear in texts p and q. I think this is a misleading explanation. What the value D(p, q) is sensitive to, is relative frequency of the words within text p and within in text q. Two texts may not exhibit one single word in common, and at the same time have a very low value of D(p, q), indicating low differential symbolic information. On the other hand, if a text p is written using an alternate alphabet and the resulting text is named q, the resulting D(p, q) value should be 0 since both text are essentially the same, in spite of the fact p and q may not share a single word, in the writing sense. Am I right? Or, am I not understanding this at all?
I like the idea of evaluating the dynamics of the lexical by using D(t, t-1). Simple, neat and powerful.
Line 367- 368. The authors present the “cut off” technique as a solution to solve the problem of the influence of the sample size over the value (p, q). This is something I do not quite understand. By cutting of the sample sizes, there cannot exist influence of sample size because there is not size difference any more. Another thing I do not completely understand is the difference between the “Litmus Test “and the “Cut off” of the sample. Clarifying these aspects would improve the manuscript.
I would have liked to see a deeper analysis on the evolution of entropy and divergence over time. But this is only a desire due to curiosity. Not showing it here it does not affect the quality of the paper because the authors study here the methods to measure entropy and divergence, and not the evolution of German language.
Final comments.
The study confirms a relevant incidence of the sample size over the computed entropy. Something that has been reported before. Yet, I think the method developed to measure this dependence has an important value and will considerably help in the search for unbiased methods to use Entropy and Entropy Divergence as comparator proxies between any two systems descriptions.
The way the authors managed to control such a voluminous data is impressive. The ingenious experiment design, aiming to avoid biased experimental conditions, is also worth of merit.
The conclusions indicate a relevant influence over text length over computed entropy and divergence. However, for most cases this influence appears in a monotonic direction, leaving the idea that in the future, some criterion could be developed to compensate for these effects.
The writing is appropriated and, in spite of the intricate the experiment and theme is by itself, the text well organized and if read carefully, can be followed. I definably recommend this article for publication in Entropy,
Gerardo Febres

Author Response
please see the attached pdf file.

Round 2
Reviewer 1 Report
I thank the authors for addressing all issues that were raised by the reviewers.
The author's revisions have lead to a much improved manuscript.
Therefore, I now recommend publication of the revised manuscript.
Reviewer 2 Report
It was interesting to re-read the article. I thank you the authors for their consideration of my comments. I think the paper could be accepted as is.